# Machinery-Induced Damage to Soil and Remaining Forest Stands—Case Study from Slovakia

**Zuzana Dudáková (Allmanová) [1,\*], Michal Allman [1], Ján Merganič [1] and Katarína Merganičová [2,3]**

[1] Department of Forest Harvesting, Logistics and Ameliorations, Faculty of Forestry, Technical University in Zvolen, T.G. Masaryka 24, 960 01 Zvolen, Slovakia; michal.allman@tuzvo.sk (M.A.); merganic@tuzvo.sk (J.M.)

[2] Faculty of Forestry and Wood Sciences, Czech University of Life Sciences Prague, Kamýcká 129, 6-Suchdol, 16500 Praha, Czech Republic; merganicova@fld.czu.cz

[3] Department of Biodiversity of Ecosystems and Landscape, Slovak Academy of Sciences, Štefánikova 3, P.O. Box 25, 814 99 Bratislava, Slovakia

\* Correspondence: zuzana.allmanova@gmail.com

**Abstract:** The paper deals with the damage of the remaining stand and soil caused by harvesting using three ground-based forest operations methods (harvester-forwarder/cable skidder/animal-tractor). It compares the impact of the most common harvesting technologies applied in Slovakia and in Central Europe and thus contributes with valuable information to the knowledge on the suitability of their application in forests stands dominated by broadleaved tree species. Harvesting was performed in five forest stands located at the University Forest Enterprise of Technical University in Zvolen in central Slovakia from August to October 2019. Damage to remaining trees was assessed from the point of its size, type, and position of damage along stem. We expected lower damage of remaining trees in stands where harvesters were used because of the applied cut-to-length short wood system and fully mechanized harvesting system. In addition, we examined soil bulk density and soil moisture content in ruts, space between ruts, and in undisturbed stand to reveal the impact of harvesting machinery on soil. We expected greater soil bulk densities and lower soil moisture content in these stands due to the greatest weight of harvesters and in ruts created by machinery compared with undisturbed stand soil. The highest percentage of damaged remaining trees equal to 20.47% and 23.36% was recorded for harvester forest operations, followed by skidder (19.44%) and animal forest operations with 19.86% and 14.47%. Factorial ANOVA confirmed significant higher soil compaction in stands where harvesters were used (higer bulk density) than in stands where skidding was performed with the skidder and animal power. Higher soil moisture content was recorded in ruts created by harvesters and the skidder. The lowest soil moisture content was in undisturbed stands irrespective of the applied forest operation method.

**Keywords:** forest operations; skidder; harvester; animal technology; wound; bulk density

## 1. Introduction

Research on the damage caused by forest operations to the remaining trees and soils in forest stands started at the beginning of the twentieth century and is becoming more important as the use of mechanized wood harvesting has been increasing since then [1]. The main impacts of logging on forests include damage to the residual stand, natural regeneration and soil surface [2]. The frequency, type, location, and patterns of damage to residual trees vary with stand characteristics, applied harvesting system, forest operations, and planning [3,4] describe the following types of residual stand damages: root abrasion and breakage, bole wounds, broken branches and crown damages. The most common

damage to residual trees is bole wounds including scarring (bark removal with sapwood exposure) and gouging (removal of wood fibres with the scar). Physical damage to tree roots and boles can result in loss of tree vigour and reduce tree growth and timber value [5]. Further, damaged trees are more susceptible to fungal infections [6]. Each harvesting system can cause distinctive damage to remaining crop trees. These damages can be caused by both felling and extracting, regardless of the technological level of machines or experience of operators. Extracting usually causes more severe damage to the remaining trees, especially when winching and skidding are applied [7]. A number of authors have dealt with the assessment of stand damage by harvesting forest operations [3,8,9].

Each harvesting system can cause soil degradation in forest ecosystems [10]. Soil as a critical element for forest sustainable management is a relatively nonrenewable natural resource [11]. Damage to forest soil can be characterised mainly by rutting, soil displacement and soil compaction caused by passage of machinery or load [12,13]. Soil compaction is in fact one of the main causes of soil degradation because it reduces soil production capacity, and leads to degradation processes, such as erosion or floods [13]. Intensity of soil compaction depends mainly on soil physical properties, followed by the load weight, terrain slope, used forest operations etc. [14]. Soil compaction causes changes in soil structure, disturbance of soil aggregates, reduction of porosity [10,15,16], and increase of bulk density [7,17]. Reduction of the space between aggregates reduces soil water content [18]. The impact of compaction varies depending on many factors such as number of passages, used harvesting machines, skid trail slope, site characteristics, production season [10]. The number and the frequency of machine passages on soil has a substantial effect on damage type [19].

The objective of this research was to compare the damage to the residual stand and soil by three different ground-based yarding methods: harvester-forwarder/cable skidder/animal-tractor. We hypothesized lower residual stand damages by cut-to-length logging (CTL) technologies because they are based on a short wood and a fully mechanized harvesting system. We examined physical soil parameters (soil bulk density and soil moisture content) in undisturbed stand parts and ruts after the application of the above mentioned three forest operation methods. We expected higher bulk densities in ruts compared with undisturbed stand soils due to the passage of forestry machinery, and higher bulk densities in stands where harvesters were applied because of their highest weight. From the point of soil moisture content, we expected greater values in undisturbed stand soils compared to the ruts.

## 2. Material and Methods

Measurements were performed at the University Forest School Enterprise of Technical University in Zvolen (UFSE), (48°37′08.2″ N 19°03′24.5″ E), the area of which is situated in a central part of Slovakia near (3 km) Zvolen. Brown forest soils covering almost 85% of forest area are most frequent at UFSE, although at lower elevations illimerised soils can also be found. Annual precipitation total fluctuates between 600 and 1000 mm, and mean annual temperature from 4 to 8 °C (Table 1). In the stands, where regeneration felling according to the shelterwood system was performed, forest operations were performed with skidders and harvester. In the thinned stands with age above 50 years forest operations were performed with animal power. Felling in stands was conducted between August and October 2019. We examined soil damage and residual stand damage at places directly affected by forest operations performed by the three technologies, namely skidder, animal, and harvester, and compared it with the situation in the remaining forest stands. In two forest stands (366a, and 366c), planned felling was performed with John Deere 1270 D harvester and John Deere 1110 E forwarder. The harvester produced assortments in 4 m lengths (short wood system). In two other forest stands (566 and 537), skidding was performed with a moderately heavy horse in combination with a universal wheeled tractor Zetor 7245 (long wood system). Skidding in the stand 507 was performed with HSM 805 HD (long wood system) (Table 2).

**Table 1.** Overview of basic stand characteristics before felling.

| Forest Stand | 366a | 366c | 507 | 537 | 566 |
|---|---|---|---|---|---|
| Technology | John Deere 1270D CTL tech. | John Deere 1270D CTL tech. | HSM805HD skidder | Horse + Zetor 7245 Animal tech. | Horse + Zetor 7245 Animal tech. |
| Age (years) | 100 | 100 | 115 | 70 | 65 |
| Area (ha) | 4.36 | 4.51 | 7.53 | 4.45 | 5.52 |
| Slope (%) | 15 | 15 | 10 | 40 | 70 |
| Aspect | NE | E | SE | SE | SE |
| Stocking | 0.84 | 0.99 | 0.73 | 0.80 | 0.80 |
| Tree species (%) | Sessile oak 92; Pine 6; Hornbeam 2 | Sessile oak 98; Beech 1; Hornbeam 1 | Sessile oak 67; Pine 22; Beech 11 | Beech 70; Fir 20; Spruce 10 | Beech 50; Sessile oak 25; Hornbeam 25 |
| Mean stem volume ($m^3$) | Sessile oak 0.84; Pine 1.63; Hornbeam 0.48 | Sessile oak 0.87; Beech 0.64; Hornbeam 0.17 | Sessile oak 1.33; Pine 1.29; Beech 1.81 | Beech 0.86; Fir 1.78; Spruce 1.50 | Beech 0.51; Sessile oak 0.37; Hornbeam 0.26 |
| Soil type | Ilimerised soil | Ilimerised soil | Ilimerised soil | Brown forest soil | Brown forest soil |
| Number of sample plots | 6 | 6 | 8 | 6 | 7 |
| Felled volume ($m^3$) | 238 | 174 | 382 | 164 | 123 |

**Table 2.** Basic information on applied technologies.

| Machine Type | John Deere 1270D | John Deere 1110 E | HSM 805 HD | Zetor 7245 |
|---|---|---|---|---|
| Dimensions width/length/height (mm) | 2766/11600/3850 | 2700/9820/3870 | 2400/5800/3200 | 2260/4530/2780 |
| Weight (kg) | 17,499 | 17,300 | 9800 | 4100 |
| Engine | John Deere 6090HTJ | John Deere 6068HTJ | Volvo Penta, four-cylinder | Z 7201 |
| Power (kW) | 160 | 136 | 129 | 46 |
| Maximum speed (km/hour) | 25 | - | - | 25 |
| Winch | - | - | Double drum ADLER HY 20SG | Single drum |
| Pulling force (kN) | - | - | 2 × 100 kN | 30.49 kN |
| Front tyres | 600 × 26.5, 20 PR Forest King F NK | 710 × 26.5−20 | Nokian Forest King TRS LS-2 23.1−26 | 9.5−24 |
| Back tyres | 600 × 34, 14 PR Forest King F NK | 710 × 26.5−20 | Nokian Forest King TRS LS-2 23.1−26 | 18.4−26 |

## 2.1. Assessment of Stand Damage

To assess the damage level of the remaining stands and soil after felling, square sample plots of 20 × 20 m size were established in individual stands. [20] stated that the size of the statistical sample should be sufficient if sample plots cover 10% of the total stand area in stands up to 50,000 $m^2$ and 5% of the total stand area in stands larger than 50,000 $m^2$ (Figure 1). Several authors determine the extent of measurements with regard to the size of the examined stand [21–24]. Based on the forest stand area and nomogram (Figure 1), we determined the required number of sample plots (Figure 2). At each sample plot, we assessed felling intensity, intensity of damage to the remaining stand, position of wounds along stem and wound size. Hence, at each sample plot we recorded numbers of felled and remaining trees, from which we could calculate intensity of felling treatments. Felling intensity is one of significant factors, which affects the damage intensity of the remaining stand. Some authors

state that the percentage of stand damage increases with the increasing harvesting intensity [25–27]. The number of felled trees was determined by counting all fresh stumps.

The percentage of damage of residual trees is one of indicators, which determine the level of damage under specific forest operations. It can be quantified as a ratio of the number of damaged trees to the total number of trees that remained in the stand after felling. This indicator is used to compare damage between individual stands and harvesting technologies applied in forest operations [28].

For the assessment of wound position along the stem we applied the classification according to [29], who specified four categories based on the tree parts: roots, buttress roots, stems at a height from 0.3 m to 1 m, stems above 1 m height (Table 3).

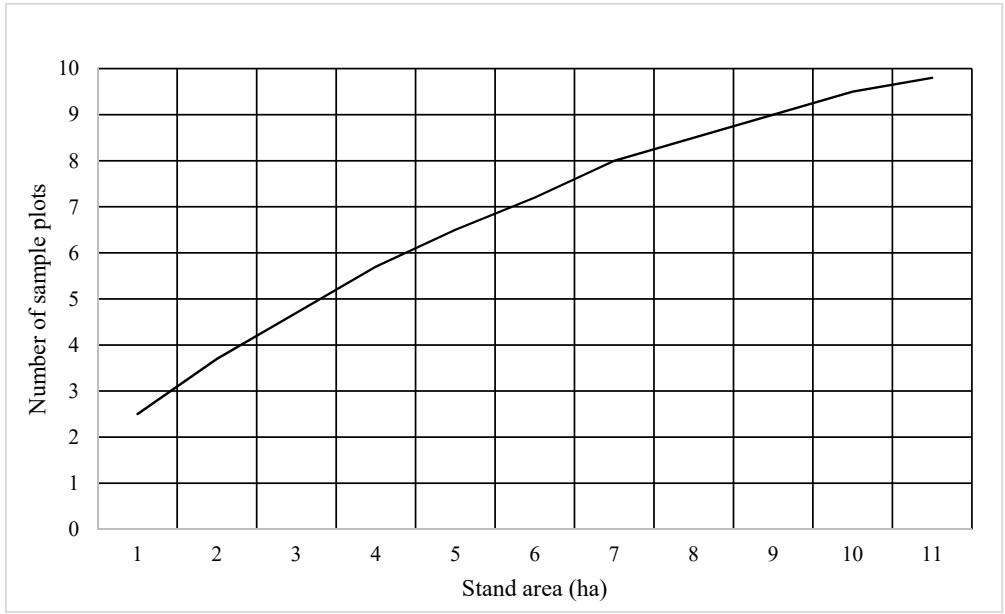

**Figure 1.** A nomogram designed to determine the number of sample plots in a forest stand [20].

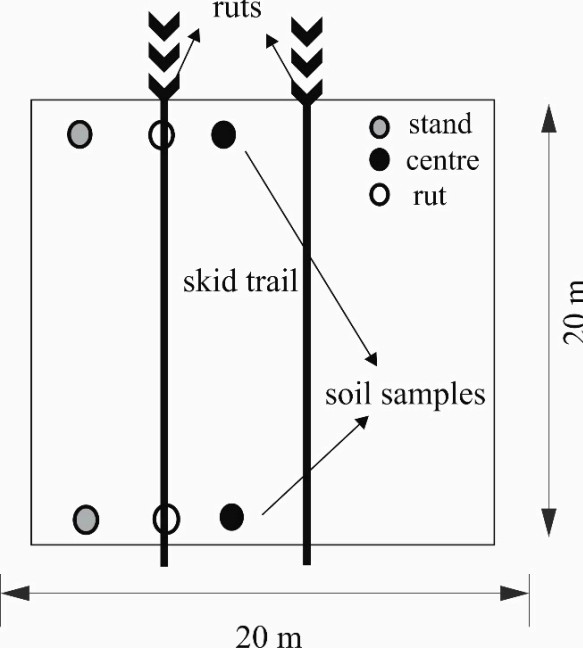

**Figure 2.** Shape and dimensions of sample plots, and location of soil sampling points (stand, centre = space between ruts, ruts).

**Table 3.** Classification of damage location on the tree [29].

| Damage Location | Characteristics |
|---|---|
| Root | Root damage (aboveground) at a distance of 0.21 to 1.0 m from the stem |
| Buttress root | Damage of the butt part of a stem at a distance of maximum 0.2 m from the stem and to the height of 0.3 m on stem |
| Stem | Stem damage at a height between 0.3–1.0 m |
| Stem | Stem damage at a height above 1 m |

At each sample plot we measured the dimensions of individual wounds with a meter and calculated their areas in $cm^2$. Wounds smaller than $10\,cm^2$ were not recorded. Afterwards, we classified wounds into individual categories (Table 4) based on the classification scheme of [29]. We also assessed the intensity of stem damage. Damage intensity was determined based on the classification scheme by [30] presented in Table 5.

**Table 4.** Classification of wounds based on the wound size [29].

| Wound Category | Damage Size ($cm^2$) | Characteristic |
|---|---|---|
| 0 | <10 | meaningless |
| 1 | 11–50 | very small |
| 2 | 51–100 | little |
| 3 | 101–200 | medium size |
| 4 | 201–300 | large |
| 5 | >300 | very large |
| 6 | Root rupture–breakage | destructive |

**Table 5.** Classification of damage intensity [30].

| Damage Intensity Class | Damage Characteristics |
|---|---|
| 1. The top layer of bark is damaged | The outer bark is damaged, cambium is undamaged, the tree reacts with low resin outflow, low risk of fungal infection |
| 2. Bark crushed (wrinkled) | Bark is wrinkled, but holds on a stem, fungal infection risk is low |
| 3. Wood exposed but undamaged | Bark is peeled off, wood is exposed but undamaged, fungal infection risk is moderate |
| 4. Wood exposed and slightly damaged | Bark is peeled off, wood is exposed and slightly damaged, high risk of fungal infection |
| 5. Wood exposed, and heavily damaged | Bark is peeled off, wood is exposed and heavily damaged, risk of fungal infection is very high |

*2.2. Assessment of Soil Damage*

To determine the changes in soil and particularly the level of soil compaction after the passage of forest machinery we took soil samples from the stands. At each sample plot, we took two samples from intact soil unaffected by forest operations, two samples from ruts, and two samples from the space between the ruts. Location of soil sampling points is presented in Figure 2. Our aim was to compare the bulk density of individual sampling points and to reveal the increase in soil bulk density due to soil compaction caused by machinery passage. Soil samples with volume of $100\,cm^3$ were extracted with an Eijkelkamp soil column cylinder auger set. Soil samples were subsequently weighed in laboratory. We recorded the weight of fresh samples prior to drying, and the weight of dried samples. Samples were

dried at a temperature of 105 °C for 24 h. Relative soil moisture was determined based on the gravimetric soil water content that was calculated from the weights of fresh and dried soil samples using the following equation: Soil moisture (%) = (Fresh soil weight − Dry soil weight)/Dry soil weight * 100.

## 3. Results

In the first part of our data analysis we dealt with the assessment of stand damage in all stands and the comparison of different forest operations from the point of their impact on the remaining forest stand. The calculated values for individual forest stands are shown in Table 6.

**Table 6.** Overview of stand damage results in individual forest stands.

| Stand ID | 366a | 366c | 507 | 537 | 566 |
|---|---|---|---|---|---|
| Number of assessed trees at sample plots | 127 | 137 | 144 | 141 | 159 |
| Number of damaged trees at sample plots | 26 | 32 | 28 | 28 | 23 |
| Damage intensity (%) | 20.47 | 23.36 | 19.44 | 19.86 | 14.47 |
| Felling intensity (%) | 18.59 | 16.96 | 18.18 | 18.02 | 16.32 |
| Sum of wound areas (cm$^2$) | 7335 | 8630 | 6830 | 7190 | 4790 |
| Mean wound area/classification according to MENG | 222.27 large | 200.69 medium size | 145.32 medium size | 194.32 medium size | 171 medium size |

The comparison of measured values revealed the highest damage intensity in the case of harvesters (20.47% and 23.36%), while the lowest damage intensity was observed in the forest stand 566, where animal power was used (14.47%). Moderate values of damage intensity were recorded for skidders (19.44%).

From the point of total wound area, the largest sum was recorded in the forest stand 366a (8630 cm$^2$), where harvesters were applied. The smallest sum of wound areas was revealed in the stand 566 (4790 cm$^2$), where skidding was performed with animal power. However, in the stand 537, where a horse was also used, the total damaged area of wounds on trees was greater (7190 cm$^2$). From the point of mean wound area, the greatest mean values were recorded in the forest stands, where harvesters were applied (222.27 cm$^2$ and 200.69 cm$^2$). The smallest mean wound areas were revealed in the forest stand, where the skidder was used (145.32 cm$^2$). Although all forest stands were prevailingly composed of broadleaved tree species, they differed in their dominant tree species (Table 1). In stands 366a and 366c, where harvesters were applied, and in stand 507 Sessile oak dominated, while in stands 537 and 566 beech was the dominant tree species (Table 1). Oaks have thicker outer bark than beech or some coniferous species, e.g., Norway spruce or Silver fir that occurred in stand 537. Hence, from the point of tree species composition we could expect greater damage in stands dominated by species with thinner bark. In spite of that, the greatest damage was observed in stands dominated by oak, where forest operations were performed with harvesters.

Next, we dealt with the location of wounds along the stem (Table 3). The results of the comparison between individual forest operations technologies are presented in Figure 3.

In Figure 3 we can see that the wounds most frequently occurred on buttress roots. In the case of harvester technologies, 51% of wounds were observed on buttress roots. The smallest proportion of wounds was recorded on stems at a height above 1 m. These wounds mainly occurred during tree felling. Roots were most damaged by skidders, as in stands where they were applied 16.1% of all wounds were observed on roots.

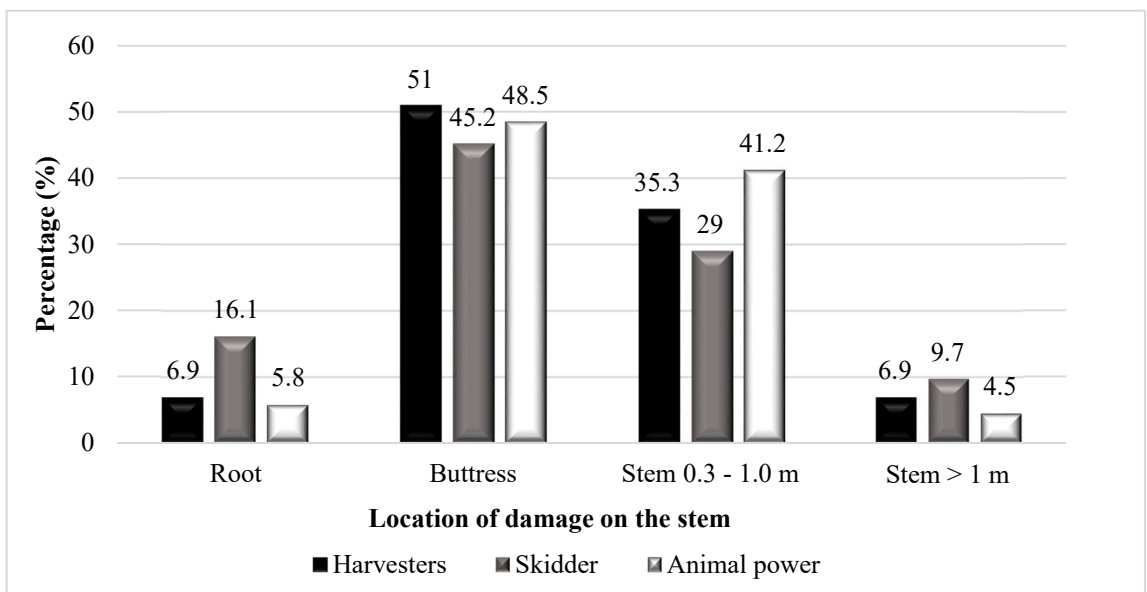

**Figure 3.** Percentage proportion of wounds at different locations along the stem.

The level of damage was assessed based on Table 5. All wounds at sample plots were classified into one class from five damage classes (Figure 4).

The results revealed that wounds were most frequently classified into damage class 3, i.e., wood exposed but undamaged (Table 5). The highest proportion of the wounds in this damage class (79.8%) was recorded for animal power, while the lowest proportion (56.3%) was observed in the case of harvesters. The highest damage class 5, i.e., wood damaged and heavily damaged, was observed only in forest stands, where harvesters were applied (2.7% of all wounds).

From the point of soil damage, we focused on the comparison of bulk density values. We were expecting higher values of bulk density in ruts than in untouched parts inside stands, and moderate values of bulk density in spaces between ruts. The results are presented in Table 7.

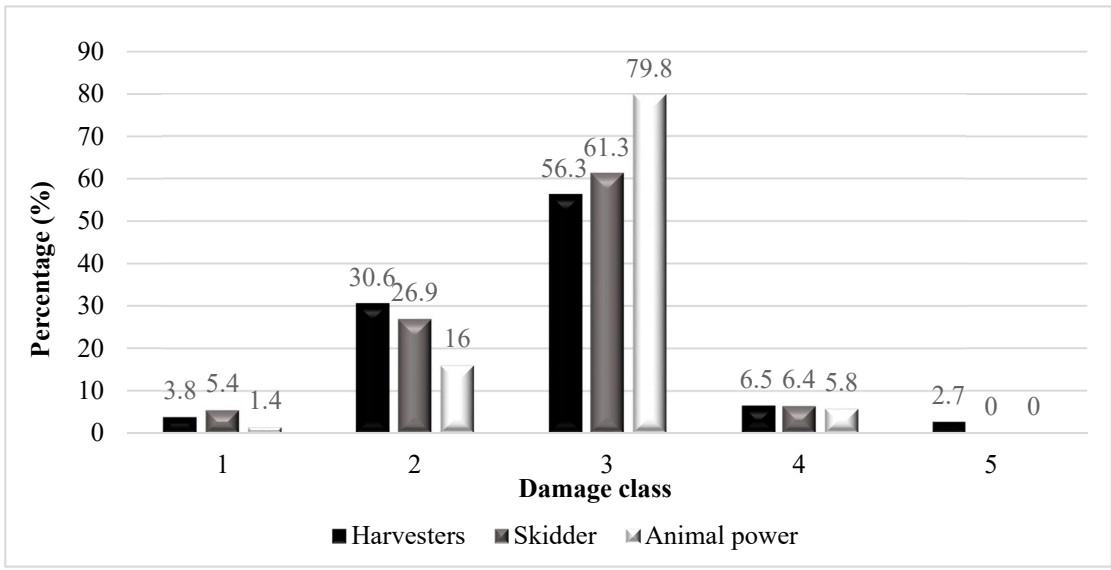

**Figure 4.** Relative distribution of wounds in individual damage classes (Table 5) with regard to individual applied technologies.

**Table 7.** Comparison of bulk density and soil moisture measured in stands, ruts, and in spaces between ruts (centre).

| Stand | 366a | 366c | 507 | 537 | 566 |
|---|---|---|---|---|---|
| Bulk density stand (g·cm$^{-3}$) | 1.15 | 1.10 | 0.95 | 0.89 | 0.91 |
| Bulk density rut (g·cm$^{-3}$) | 1.17 | 1.12 | 1.06 | 1.03 | 1.01 |
| Bulk density centre (g·cm$^{-3}$) | 1.20 | 1.08 | 1.03 | 0.96 | 0.95 |
| Moisture stand (%) | 10.47 | 11.62 | 20.14 | 23.59 | 19.51 |
| Moisture rut (%) | 15.29 | 17.60 | 28.40 | 25.71 | 21.38 |
| Moisture centre (%) | 11.69 | 13.19 | 21.70 | 25.82 | 22.19 |

The results confirmed the differences in bulk density. The greatest difference in bulk density of 0.14 g·cm$^{-3}$ was observed between the stand and the rut in the forest stand 537, where a combination of animal power and the universal wheeled tractor was applied. In the forest stand 566, the difference between the stand and the rut was 0.1 g·cm$^{-3}$. The smallest differences were revealed for harvesters. In the forest stand 366c, the difference between the stand and the rut was 0.02 g·cm$^{-3}$. The highest mean bulk densities in the middle between the ruts (centre) and in the rut equal to 1.20 and 1.17 g·cm$^{-3}$, respectively, were recorded in the forest stand 366a. The difference between the mean values representing the stand and the middle space between the ruts in 366a was 0.05 g·cm$^{-3}$. These minimum differences were partly caused by the harvest residuals occurring at some parts of the skid trail used by the harvester. In the case of the skidder technology, the greatest difference of 0.11 g·cm$^{-3}$ between the stand and the rut was found in the forest stand 507.

Factorial ANOVA (Figure 5) was used to compare soil characteristics between forest stands and measurement locations. The results showed significant differences in the level of soil compaction between individual stands (F = 14.95; $p$ = 0.00) but not between the position of measurement (F = 2.95; $p$ = 0.054) in individual stands. Differences between stands and position of measurements (stand*locations) were significant (F = 2.26; $p$ = 0.0251), (Figure 5).

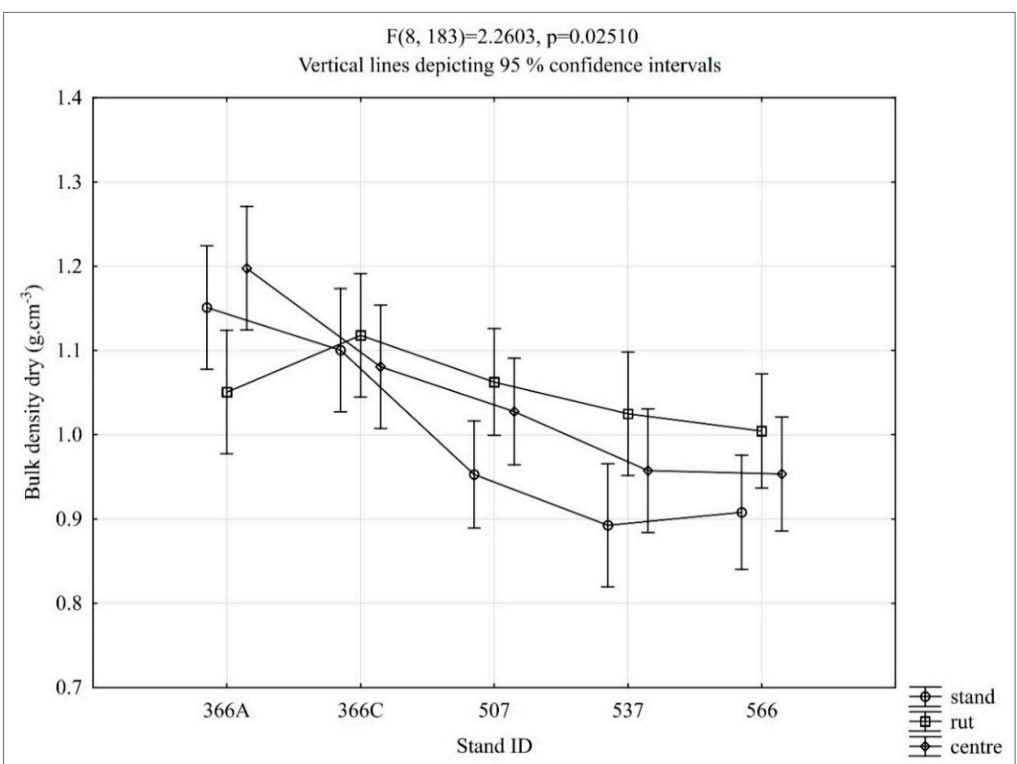

**Figure 5.** Analysis of differences in average bulk density of dried soil samples between individual stands and measurement locations.

Duncan's test was used to analyse differences between dried soil samples from ruts. The results of Duncan's test confirmed the significant difference between stands where harvesters were used and other stands. Bulk densities of soil in stands with harvesters were in the range from 1.12 g·cm$^{-3}$ to 1.17 g·cm$^{-3}$, while in stands where skidding was performed with skidders or a combination animal + tractor they were only 1.06 g·cm$^{-3}$ or 1.01–1.03 g·cm$^{-3}$, respectively. These results indicate that harvesters caused higher soil compaction than skidders or combined animal power and tractor. Duncan's test did not confirm a significant difference between skidder and animal power + tractor (Table 8).

**Table 8.** Duncan's test of soil bulk density in ruts.

| | Duncan's Test, Average Dry Bulk Density (g·cm$^{-3}$) Approximate Likelihood of Post Hoc Tests Error: Between Groups = 0.01783, Degrees of Freedom = 193.00 | | | | |
|---|---|---|---|---|---|
| **Stand** | **366a** **1.1331** | **366c** **1.0998** | **507** **1.0143** | **537** **0.95824** | **566** **0.95525** |
| 366A | | $2.70 \times 10^{-1}$ | $1.33 \times 10^{-4}$ * | $3.00 \times 10^{-6}$ * | $4.00 \times 10^{-6}$ * |
| 366C | $2.70 \times 10^{-1}$ | | $4.72 \times 10^{-3}$ | $1.50 \times 10^{-5}$ * | $6.00 \times 10^{-6}$ * |
| 507 | $1.33 \times 10^{-4}$ * | $4.72 \times 10^{-3}$ * | | $6.33 \times 10^{-2}$ | $6.37 \times 10^{-2}$ |
| 537 | $3.00 \times 10^{-6}$ * | $1.50 \times 10^{-5}$ * | $6.33 \times 10^{-2}$ | | $9.21 \times 10^{-1}$ |
| 566 | $4.00 \times 10^{-6}$ * | $6.00 \times 10^{-6}$ * | $3.37 \times 10^{-2}$ | $9.21 \times 10^{-1}$ | |

* Significant difference.

The analysis of soil moisture revealed the greatest differences between sampling locations for skidder. In the stand 507, the soil moisture inside the stand was 20.14%, while in the rut and in the space between the ruts it was 28.40% and 21.70%, respectively. The difference in the soil moisture between the stand and the rut was 8.26%. On the contrary, the smallest differences were observed in the stand, where a combination of a horse and a universal wheeled tractor was used. Here the lowest soil moisture was found in the stand. The differences of the soil moisture in the stand from the centre and the rut were 0.81 and 0.11%, respectively. These minimum differences resulted mainly from dragging stems along the skid trail. The overall lowest values of soil moisture were measured for harvesters. The differences between the mean soil moisture in the stand and in the rut were 4.82% and 5.98% in the stands 366a and 366c, respectively. Factorial ANOVA (Figure 6) revealed significant differences in soil moisture content between individual stands (F = 36.441; $p$ = 0.00) as well as between measurement locations: stand, rut, centre in individual stands (F = 10.777; $p$ = 0.00). Differences between stands and their position of measurements (stand * locations) were not significant (F = 1.27; $p$ = 0.257), (Figure 6). The results of Duncan's test confirmed significant differences between stands with the skidder and animal skidding and stands with harvester logging. The highest moisture content of 28.40% was revealed in the stand with skidder, followed by tractor + animal in the range from 21.38% to 25.71%. The lowest soil moisture content in ruts (between 15.29% and 17.60%) was found for harvesters (Table 9).

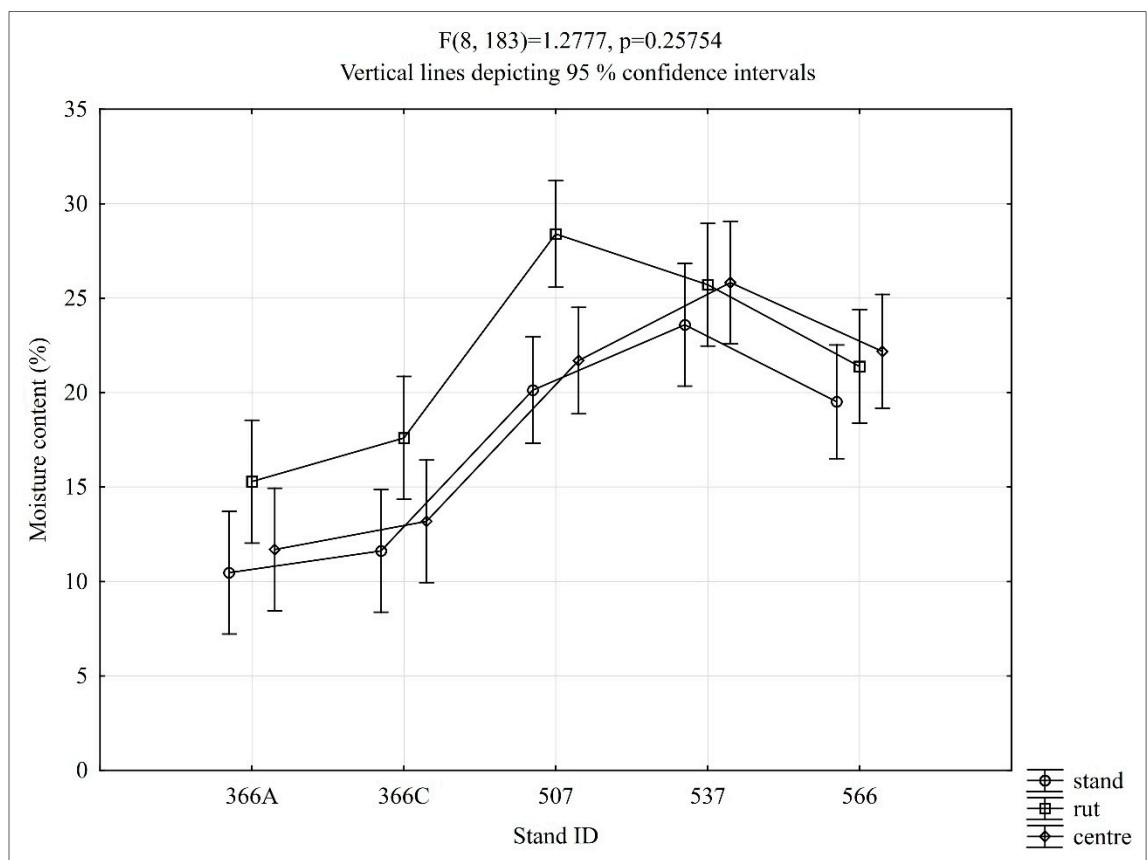

**Figure 6.** Average soil moisture values in individual stands and measurement locations.

**Table 9.** Duncan's test of soil moisture content in ruts after compaction.

| | Duncan's Test, Soil Moisture Content (%) Approximate Likelihood of Post Hoc Tests Error: Between Groups = 36.525, Degrees of Freedom = 193.00 | | | | |
|---|---|---|---|---|---|
| **Stand** | **366a** **12.484** | **366c** **14.135** | **507** **23.413** | **537** **25.043** | **566** **21.028** |
| 366A | | $2.27 \times 10^{-1}$ | $3.00 \times 10^{-6}$ * | $4.00 \times 10^{-6}$ * | $1.10 \times 10^{-5}$ * |
| 366C | $2.27 \times 10^{-1}$ | | $1.10 \times 10^{-5}$ * | $3.00 \times 10^{-6}$ * | $9.00 \times 10^{-6}$ * |
| 507 | $3.00 \times 10^{-6}$ * | $1.10 \times 10^{-5}$ * | | $2.33 \times 10^{-1}$ | $8.12 \times 10^{-2}$ |
| 537 | $4.00 \times 10^{-6}$ * | $3.00 \times 10^{-6}$ * | $2.33 \times 10^{-1}$ | | $4.68 \times 10^{-3}$ * |
| 566 | $1.10 \times 10^{-5}$ * | $9.00 \times 10^{-6}$ * | $8.12 \times 10^{-2}$ | $4.68 \times 10^{-3}$ * | |

* Significant difference.

## 4. Discussion

Our results revealed that the lowest mean relative damage of the remaining stand equal to 17.17% was recorded for the animal power in combination with the universal wheeled tractor. Other authors recorded stand damage caused by animal power from 12.17% [31] to 37.29% [32]. In the case of forest skidders, the percentage of damage to the remaining forest stand (17.76%) was substantially lower than the value published by [33], who reported stand damage equal to 44.64%. [34] found residual stand damage by skidder 49.9%. [29] reported the maximum value of stand damage equal to 21.2% if a forest skidder and a short wood method was applied. Interesting results were presented by [35], who analysed tree damage in beech stands after felling with BOBCAT 733 and LKT 81. They analysed 1789 trees at four plots, out of which 388 trees were damaged (i.e., 21.17%). The relative damage at individual plots fluctuated from 17 to 28%. In our case, harvesters damaged the remaining stand

most with the mean stand damage of 21.92% in stands with a mean slope of 15%. According to [36], mean stand damage by harvesters is 17%. However, [37] found stand damage by harvesters equal to only 7.36%. [38] dealt with the damage of tree roots in a spruce stand after harvesting with a tracked harvester, and revealed 15.1% stand damage. From the point of wound size, we found the greatest mean wound size equal to 201 and 222 cm$^2$, which according to [29] represents "great" damage, in the stands where harvesters were used. Lower values (171 and 194 cm$^2$) were recorded in the case of animal power, while the wounds were of medium size. The smallest mean sizes of wounds equal to 145 and 161 cm$^2$, classified as wounds of medium size, were found in the stands harvested by skidders. [35] reported that wounds at heights between 30 and 100 cm represented 40% of all wounds, and their mean wound size fluctuated between 466 to 1190 cm$^2$. [35] states wounds at tree roots were from 10 to 200 cm$^2$ large in mixed beech-oak stands. [39] compared the mean size of wounds that occurred during forest operations performed by skidders and harvesters. He revealed the smallest mean sizes of 134.21 and 200 cm$^2$ in stands, where a tracked harvester was applied (spruce-thinning). This was followed by a wheeled harvester with 255.43 and 388.89 cm$^2$ large wounds (oak-thinning), and the largest wounds of 395.13 and 505.81 cm$^2$ were observed in the case of skidders (oak and pine, shelterwood) [39].

If we considered tree species composition, the lowest damage would be expected in stands composed of tree species with thick outer bark, such as oak or pine. Thicker bark should protect vascular cambium from logging injuries. Bark protection against a direct impact from a falling stem or from harvesting equipment would be expected to increase with greater impact strength and higher bark density [40]. Tree species with thinner bark such as Norway spruce and Silver fir are highly susceptible to wounds and decays [41,42]. Despite our expectations, we observed the greatest damage in oak stands, where harvesters were used. Harvesters were originally developed for forest operations in coniferous stands, but recently they have been applied also in broadleaved forests, particularly in Central Europe [43]. The effectiveness of harvester application in broadleaved forests depends on tree quality [44], which can also affect damage to remaining trees. If harvested trees have long and big crowns with numerous and thick limbs and branches, logging with a harvester can increase damage to the remaining stand. Another way to explain this phenomenon is the impact of an operator, since [45,46] showed that operator's skills can have a direct impact on the degree of damage but training (experience) [47,48] can lead to lower damages. [49] also identified the human factor as a key factor affecting damage extent.

Harvesting time within a year can also have a significant impact on stand damage. In our study, harvesting was performed from August to October, when tree species are less susceptible to damage than in early spring [50].

Soil damage intensity was analysed on the base of soil bulk density and soil moisture. In all but two forest stands we found the highest bulk density in ruts after the passage of machinery, and the lowest bulk density inside the stands. The highest value of bulk density in ruts was recorded in the forest stand 366a, where the harvester was used, but the differences between the individual sample locations were low reaching 0.05 g·cm$^{-3}$. The greatest difference in dry soil bulk density between the stand and the rut of 0.14 g·cm$^{-3}$ was observed for animal skidding, while the lowest difference equal to 0.02 g·cm$^{-3}$ was found for harvesters. Factorial ANOVA did not reveal significant differences between measurement locations, which was very "surprising" for us. One explanation of small differences in the case of skidding with animals and skidders is the fact that the load is pulled on the ground. In such a case, the space between ruts is also affected due to which soil compaction occurs not only in the ruts but also between them. [51] analysed the changes in soil bulk density in stands and ruts after the passage of universal tractors and forest skidders, and found that in both cases the bulk density in ruts increased by 0.26 g·cm$^{-3}$. Factorial ANOVA confirmed significant differences in soil compaction between stands. Soil in stands 366a and 366c, where harvesters were used, was more compacted in ruts than in other stands, where different harvesting technologies were used for forest operations. Similar results were presented by [24], who found that wheeled harvesters caused the increase of bulk density

by 35.4%, while skidders increased the bulk density by only 30.3%. [52] dealt with critical thresholds of soil damage for several soil properties, and specified the threshold bulk density, at which roots still grow, from 1.5 to 1.7 g·cm$^{-3}$. These values were not exceeded in any of the examined cases in our study.

Factorial ANOVA confirmed significant differences in soil moisture content between individual stands and between positions of measurement (stand, rut, centre in individual stands). The results of Duncan's test confirmed the difference between skidder and animal technologies versus harvesters. The moisture content was lower in ruts of stands where harvesters were used. The relationship between bulk density and soil moisture was examined by [53], who found that bulk density decreased with the increasing soil moisture. The study was performed after the passage of wheeled forwarders on soils with various soil moisture. In our case we observed greater soil moisture in stands (507, 566, 537), where skidder and animal technology with a wheeled tractor were used, in comparison to harvesters, which were used in stands with lower soil moisture and greater bulk densities.

## 5. Conclusions

Timber harvesting has a significant influence on forest environment. Its direct impacts are visible on the remaining forest stands in the form of wounds on remaining trees as well as on the soil in the form of ruts. Currently, a great number of forest harvesting technologies can be applied to perform harvesting forest operations with different impacts on remaining trees and soil. Our study revealed greatest damage in stands where shelterwood regeneration harvesting was performed by harvesters (20 and 23%), while stand damage caused by a skidder and a long wood system was lower (19%) at comparable felling intensity and stand characteristics. Animal skidding applied in thinning operations of similar intensity resulted in 14% and 19% stand damage. Forest operations have substantial negative impacts also on forest soil. Soil damage is observed via changes in physical characteristics of the soil. Common methods used to estimate soil damage severity include the measurement of soil bulk density. Our results confirmed significant differences between stands where harvesters were used and other stands. The bulk densities of soil from stands with harvesters were in the range from 1.12 g·cm$^{-3}$ to 1.17 g·cm$^{-3}$. The results confirmed our expectations that the highest values of soil bulk density were found after forest operations performed with harvesters, but the critical values of bulk density were not exceeded. Limit values for root growth vary from 1.4 g cm$^{-3}$ for clay soils to 1.8 g·cm$^{-3}$ for sand and loamy sand soils. Soil in stands where a skidder or animal power were applied, had lower soil bulk density and thus, damage to soil was not of such an extent that would limit further root growth.

**Author Contributions:** Conceptualization, Z.D. and M.A.; methodology, Z.D. and M.A.; investigation, Z.D. and M.A.; data curation, Z.D.; writing—original draft preparation, Z.D. and M.A.; writing—review and editing, Z.D., M.A., K.M., J.M.; visualization, Z.D.; project administration, J.M.; funding acquisition, J.M. All authors have read and agreed to the published version of the manuscript.

**Funding:** This work was supported by the Slovak Research and Development Agency (APVV) [grant number 18-0305] "Utilisation of progressive methods for evaluation of forest logging impacts on forest ecosystems and road network", by [grant number 15-0714] "Mitigation of climate change risk by optimization of forest harvesting scheduling", by Scientific Grant Agency VEGA [grant number 1/0241/20] "Optimization and environmental impact of logging technologies in close to nature forest management", the grant 'EVA4.0', No. CZ.02.1.01/0.0/0.0/16_019/0000803 financed by OP RDE, and the project: Scientific support of climate change adaptation in agriculture and mitigation of soil degradation (ITMS2014 + 313011W580) supported by the Integrated Infrastructure Operational Programme funded by the ERDF.

**Conflicts of Interest:** The authors declare no conflict of interest.

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
