# Peer review of "Machinery-Induced Damage to Soil and Remaining Forest Stands—Case Study from Slovakia"

_forests, doi:10.3390/f11121289_

Round 1
Reviewer 1 Report
The work concerns a important topic, which can be understood not only locally but also regionally in Europe. Unfortunately, starting from the title of the publication, through the chapters: Abstract, Keywords, Introduction, Discussion and Conclusions need to be completely redesigned.
Abstract requires a revision to improve the comprehension of the work.
The introduction is deprived of an analysis of the current literature of the examined subject.
I do not see in it the main stream which would end in the indication of the research hypothesis and the clearly demonstrated purpose of the research.
Results require reconstruction - in the part concerning damage to the remaining stand.
Lack of clearly defined objectives in the introduction means that the chapter Conclusions cannot be assessed. The research goal and the conclusions drawn from its conduct are the basis for assessing whether the work meets the conditions of a scientific publication. The conclusions described in this paper do not result from the studies presented above.
Linguistic proofreading and native work is absolutely necessary.
Line 2-5 the title of the publication must be changed. The reader gets the impression that the place where the experiment is performed is more important than the research itself. I suggest using in the name: "case study from Slovakia" or similar.
Line 15-27 Abstract should be rewritten - following the instructions in the Forests articles: “Research Highlights: Place the novelty of the content and highlight the significance of the study. Background and Objectives: Place the question addressed in a broad context and highlight the purpose of the study. Materials and Methods: Describe briefly the main methods or treatments applied, including the study population description. Results: Summarize the article's main findings. Conclusions: Indicate the main conclusions or interpretations.” The authors use the words: technology, technologies. They probably mean forest operations (which is the correct term) using various types of equipment and techniques.
Line 16 It seems that the author, when using the words: “technology, technological” means forest operations with the use of various machines and techniques. Please use the term used in publications on wood harvesting: "forest operations"
line 17 Same as above.
line 23 Same as above.
line 24 Same as above.
line 28 Keywords - they cannot contain words used in the title of the work!
line 31 Introduction should be rewritten! In the introduction, the authors indicate 7 citations from which: 1 textbook for forestry students (1999), 4 publications from Slovakia (2011-2016), one publication from Sweden and one publication from the Web of Science .... Respecting the knowledge resulting from textbook (research and lectures) by Professor Lukac (whom I had the honor of meeting in person), the reader is under the impression that no similar experiments have been conducted anywhere except Slovakia - which, unfortunately, cannot be agreed.
An analysis of only publications from "Forests" from the last 4 years shows current works in this fiel (damage to trees and soil) both in Europe and in the United States.
It is necessary to carry out an extensive analysis of the literature on the subject, taking into account global (or at least European) publications in this field from at least the last 10 years!
The lack of references to current research in the forest knowledge with particular emphasis this research in Europe and America (except for the spontaneous used Poršinsky and Arvidsson publication) means that this chapter cannot be accepted.
I do not see in it the main stream which would end in the indication of the research hypothesis and the clearly demonstrated purpose of the research.
Line 56 At what time of the year was the forest operations done? Were the works carried out at the same time on each experiment (month, part of the year, growing season)?
Line 58-60 Instead of using geographical names (known mainly in Slovakia), please write in which part of Slovakia the experiment was carried out and give the latitude and longitude.
Line 62-63 There is no need to describe that any item is in the table. Please, after the description in the text, insert the reference to the table.
Line 65-67 Same as in line 17
Line 71 And what length of assortments were there in other (except CTL) forest operations?
Line 73 Felling in the stand 507 with HSM 805 HD? Maybe skidding?
Line 74 More detailed information on parameters of applied technologies is in Table 2. Same error as in Line 62-63. It is enough to put (Table 2) after the word Zetor 7245.
Table 1 Line 3 (Forest management unit) and Line 4 (management unit) in table 1. Why insert if the descriptions for all columns are identical?
Linie 79-80 You write - I quote: “ A number of national and international authors determine the extent of measurements with regard to the size of the examined stand.” And…you don't cite any of this great number of authors. You only refer to Fig. 1, which refers to the [9]- work of Ulrich et al 2003, which is additionally poorly described in the reference list. Absolutely unacceptable simplification!
Linie 100-104 The data in Fig 3 and Tab 3 show the same! So why do the authors use both?
Linie 124 The results are presented in a disorderly fashion. Table 6, similarly to the chapter "material and methods", maintains the order of the research areas in the following system: 507,537,566,366a, 366c. On the other hand, the next Figure 4, for reasons known only to the author of the work, changes the order of the research plots to 366a, 366c, 507, 537, 566. This causes the results to become illegible and difficult to interpret for a potential reader!
Linie 125-161 Results concerning damage to the remaining trees should be verified. The authors compare the wound surfaces on trees with different bark thickness, assigning different forest operations to them (for example: oak - CTL, beech - chainsow and tractor / horse). Both the results and the discussion did not pay attention to this fact - and it may affect the frequency and area of damage and erroneously "burden" the type of forest operations.
Linie 218-245 It requires correction and taking into account in the discussion that trees with different bark thicknesses were analyzed.
Linie 277-286 These conclusions do not emerge from the studies presented above.
Reviewer 2 Report
This paper deserves publication in FORESTS provided some minor improvements could be performed:
- Lines 79-81, “A number of national and international authors determine the extent of measurements with regard to the size of the examined stand”: At least one reference is needed here.
- Lines 80-81, “Based on the 80 forest stand area and nomogram (Fig. 1)”: What is the source for this nomogram?
- Lines 86-87, “it is generally known that the percentage of stand damage increases with the increasing felling intensity”: Reference
- Lines 87-89, “The percentage of damage is one of primary and most important indicators, which determine the level of damage under specific technology”: Reference?
* Line 135, “Figure 4. Damage intensity for individual stands with 95% confidence intervals”: Is it necessary to show these confidence intervals, since it seems evident that there was no statistical treatment of the data?
